# In Vitro Evaluation of the Antioxidant Activity and Chemopreventive Potential in Human Breast Cancer Cell Lines of the Standardized Extract Obtained from the Aerial Parts of Zigzag Clover (*Trifolium medium* L.)

**DOI:** 10.3390/ph15060699

**Published:** 2022-06-02

**Authors:** Grażyna Zgórka, Magdalena Maciejewska-Turska, Anna Makuch-Kocka, Tomasz Plech

**Affiliations:** 1Department of Pharmacognosy with the Medicinal Plant Garden, Faculty of Pharmacy, Medical University of Lublin, 1 Chodźki Str., 20-093 Lublin, Poland; magdalena.maciejewska@umlub.pl; 2Department of Pharmacology, Chair of Pharmacology and Biology, Faculty of Health Sciences, Medical University of Lublin, 20-093 Lublin, Poland; anna.makuch@umlub.pl (A.M.-K.); tomasz.plech@umlub.pl (T.P.)

**Keywords:** *Trifolium medium* L., isoflavones, chemoprevention, cytostatic activity, MCF-7, MDA-MB-231

## Abstract

The aboveground parts of *Trifolium medium* L. (zigzag clover), a little-known representative of the family Fabaceae, collected during flowering in a wild stand (Sławin-Szerokie district, Lublin, Poland), were used in this study. Our previous investigations confirmed the higher content of phytoestrogenic isoflavones (especially biochanin A and formononetin derivatives) in *T. medium* compared to the closely related medicinal plant *T. pratense* (red clover) and the involvement of these compounds in anti-osteoporotic effects in ovariectomized female rats. The current study focused on evaluating other antibiodegenerative (antioxidant, chemopreventive, and cytostatic) effects for the lyophilisate (TML) obtained from wild zigzag clover. For this purpose, efficient ultrasound-assisted extraction (UAE) was employed, followed by vacuum drying and phytochemical standardization using a newly developed reversed-phase high-performance liquid chromatography (RP-LC) coupled with a PDA detection. Malonylglycosides of biochanin A and formononetin were the predominant compounds and were found to contribute more than 54% to the total isoflavone content determined in the standardized extract of zigzag clover. The antioxidant potential of TML was examined in vitro using the Folin–Ciocalteu and cupric ion-reducing (CUPRAC) methods in addition to the free radical (DPPH^•^ and ABTS^•+^) scavenging assays. The cytotoxic effects of TML, formononetin, and ononin were evaluated on MCF-7 (estrogen-dependent) and MDA-MB-231 (estrogen-independent) human breast cancer cell lines using the MTT assay. The important role of malonyl isoflavone derivatives has been indicated both in chemoprevention and potential cytotoxic effects of TML against certain types of breast cancer.

## 1. Introduction

For many years, soybean preparations characterized by the high content of genistein and daidzein derivatives have been considered the richest herbal source of isoflavones [1]. However, over the past two decades, red clover (*Trifolium pratense* L.), a member of the same botanical family, Fabaceae, has begun to draw the attention of researchers because of the significant content of these specialized plant metabolites documented in above-ground green parts of this taxon. Compared to soybean, red clover isoflavones showed a slightly different qualitative profile with a predominance of biochanin A and formononetin and their glycosides [2]. Rich evidence from epidemiological and clinical studies also indicated a direct relationship between high consumption of isoflavone-rich food (e.g., soy-based products), dietary supplements, and herbal medicinal products and a reported, especially in Asian populations, significant reduction in the development of osteoporosis and a lowered incidence of menopausal complaints and hormone-related cancers [3]. Moreover, dietary intake of natural antioxidants has gained widespread approval among both nutritionists and physicians as a new chemopreventive strategy to reduce the risk of carcinogenesis, particularly breast cancer, which, according to the current data published by the WHO, is one of the most commonly diagnosed malignancies in women [4,5]. It is noteworthy that early chemoprevention can reduce ROS-induced oxidative damage to key cellular macromolecules such as nucleic acids, thereby stabilizing the genome and reducing the probability of initiating the multistep process of carcinogenesis [6]. Phytochemicals, especially phenolic compounds, are known to exert anticancer effects through various molecular pathways including specialized ROS scavenging mechanisms, lowering elevated redox balance in malignant cells, modulating hormonal and enzymatic activities, and inducing DNA damage that leads to apoptosis of malignant cells [7,8]. Therefore, facing the problem of aging populations around the world, scientists recognize the need to search for new natural sources of biologically active ingredients that could counteract progressive degenerative diseases (including those related to tumor growth) and pose less risk of side effects compared to synthetic drugs [9]. As for isoflavones, it is thought that antioxidant properties may coexist with their ability to modulate factors involved in carcinogenesis. Current research indicates that these compounds may limit the development of certain types of breast and/or ovarian cancer not only through the effects exhibited through estrogen receptors (ERs) but also by reducing cellular proliferation and tumor formation via an estrogen-independent pathway [10,11]. The specific chemical structure of isoflavones enables sterical fitting into the ligand-binding ER domain and inducing estrogen-like signaling comparable to 17β-estradiol [12]. They may also exert a dual agonistic/antagonistic response, similar to selective estrogen receptor modulators (SERMs), that depends on the ER isoform (α or β) present in the target tissue [12]. Stereospecific interactions of isoflavone ligands against β-type estrogen receptors (particularly abundant in bones, the central nervous and cardiovascular systems, the prostate, the urinary bladder, the breast epithelium, and ovaries) result in activation of a cascade of endogenous cellular factors responsible for cytoprotection, anti-inflammatory effects, and chemoprevention in the aforementioned mammalian tissues and organs [13]. Results from several published mechanistic studies have also shown that overexpression of ER-β in tumor cells and/or treatment with ER-β agonists reduced tumor cell proliferation and resistance to chemotherapeutics [14]. From the perspective of our investigation, it was relevant that the enhanced chemosensitivity induced by ER-β activation has already been documented in vitro in MCF-7 luminal breast cancer cells [15]. Promising results from a clinical trial of 118 women undergoing surgery for invasive breast cancer have also been published, showing improved survival in women with ER-β expression in tumor tissue [16].

All the above reports showed multidirectional biological effects of isoflavones, resulting from both their polyphenolic structure and estrogeno-mimetic properties. In our previous studies, we determined a detailed profile of zigzag clover phenolics showing, beside predominant phytoestrogenic isoflavones, some other minor phytochemicals, including flavonols, flavanones, and caffeic acid derivatives [2]. Moreover, Zgórka [17] documented that the total content of isoflavone aglycones (formononetin and biochanin A) in *T. medium* was about three times higher than the average amount of these compounds in red clover, which is widely used in herbal medicinal products for various menopausal complaints [18,19]. Results of our current studies (not yet published) have revealed significant influence of agro-environmental conditions (plant-growth phase, insolation of the growing area, harvesting time, etc.) on the content of isoflavones in aerial parts of zigzag clover. Other researchers have also found seasonal variation in the concentration and estrogenic activity of isoflavones in inflorescences and whole green shoots of red clover during the growing season [20]. Lemežienė et al. [21] and Tsao et al. [22] additionally confirmed the variable content of red clover isoflavones depending on the aboveground part and cultivar type of this taxon, indicating the importance of genetic factors in the biosynthesis of these specialized plant metabolites.

Referring to the above reports, in this study, we focused on the preparation of a highly concentrated vacuum-dried extract (TML) from the aboveground parts of wild zigzag clover, harvested when the plant was in full flowering, and its further phytochemical standardization. We hypothesized that under these conditions we would obtain a high-quality herbal product with significant content of isoflavone phytoestrogens for further biological studies. Considering the results of our previous in vivo experiments, showing SERM-like antiosteoporotic effects of red and zigzag clover isoflavones in ovariectomized rats [23], we decided to investigate other antibiodegenerative properties of TML, including the antioxidant/chemopreventive potential in vitro, and then evaluate cytostatic effects using both the estrogen-dependent (MCF-7) and independent (MDA-MB-231) human breast cancer cell lines.

## 2. Results and Discussion

### 2.1. Preliminary Procedures Related to the Raw Plant Material Preceding Phytochemical Standardization 

Proper selection of plant material, harvesting time, and drying procedure were indicated by researchers as crucial for the quality of herbal substance and obtaining standardized plant preparation [24]. The official guidelines and regulations for quality control of herbal medicinal products during their development (e.g., including proper preparation of the raw plant material) are described in a special section of the European Pharmacopoeia relating to pharmacognostic methods [25]. Our long-term experiments (not yet published) on optimizing the harvesting time of the herbaceous parts of zigzag clover showed some variability in the content of individual isoflavone compounds depending on the plant-growth phase and vegetation period. We also observed differences in the concentration of these compounds between wild-grown plants and those whose controlled cultivation was carried out in the Medicinal Plant Garden of the Department of Pharmacognosy (Medical University of Lublin, Poland). Therefore, in the present study, we decided to use plant material obtained from a natural habitat located near Lublin (Sławin region) with a high exposure to sunlight during the summer period. 

### 2.2. Phytochemical Standardization of TML Using Coupled Chromatographic (RP-LC), Spectroscopic (PDA), and Mass Spectrometric (ESI-QTOF-MS/MS) Techniques

A new RP-LC/PDA protocol was developed as the main analytical procedure in phytochemical standardization of TML in order to identify the composition and real amounts of isoflavone constituents both in herbal substance and the lyophilisate prepared thereof. The studies included qualitative profiling and determining the content of isoflavone components that (as described below) significantly influenced the biological effects exerted by TML. Additionally, an ESI-QTOF-MS/MS procedure was used to establish the molecular structures and/or confirm the identity of isoflavones found in TML.

#### 2.2.1. Qualitative Profiling of Isoflavones in TML Using Simultaneous RP-LC/PDA and RP-LC/PDA-ESI-QTOF-MS/MS Methods

Eight major isoflavone compounds were detected in TML using the newly developed RP-LC/PDA method by relating UV spectra of analytes examined to those of standard isoflavone substances collected in the ChemStation Rev. 10.02 spectral library included in the 1100 Agilent chromatographic system. In this group, six isoflavones, namely, three aglycones (**6–8**) and their corresponding 7-*O*-glycosides (**1**, **2**, and **4**), were unambiguously identified based on the conformity of their retention times with the reference substances (Figure 1). 

Two of the detected but unidentified isoflavones (**3** and **5**) had UV-spectra very similar to ononin (**2**) and sissotrin (**4**). Therefore, based on our previous experiences [2] and the results of qualitative isoflavone profiling presented by Dulce-María et al. [26] for sprouted chickpea extracts, we suspected the presence of malonyl derivatives of ononin and sissotrin in TML. To verify the above hypothesis, the identity of all isoflavone compounds was additionally confirmed after chromatographic separation using the more sophisticated mass spectrometric (ESI-QTOF-MS/MS) method, according to the analytical procedure described in our previously published paper [2]. By comparing the *m*/*z* values obtained for precursor ions and fragmentation patterns of product ions generated by collision-induced dissociation (CID) with those of reference substances and literature data, the identity of eight isoflavone compounds was unambiguously confirmed and their molecular structures were determined as shown in Figure 2.

The molecular ions acquired at *m*/*z* 271.0600, 269.0809, and 285.0756 stood for isoflavone aglycones, namely **G**, **F**, and **B**, respectively (Figure 2). The loss of small molecules (CO, CO_2_) typical for isoflavones as well as diagnostic fragments of retro-Diels-Alder (*r*DA) cyclization at *m*/*z* 137.0650 and 152.0122 [27] were also obtained for **F** and **B**, respectively. In the case of their corresponding *O*-glucosides (**F-Gl** and **B-Gl**), the neutral loss of glucose moiety (−162 Da) from the [M+H]^+^ ions resulted in the detection of distinct fragment ions originating from the aglycone structure. Compounds showing abundant deprotonated ions at *m*/*z* 433.1129, 431.1325, and 447.1295 were unambiguously identified as **G-Gl**, **F-Gl**, and **B-Gl**, respectively. An electrospray ionization (ESI) performed in positive mode provided efficient generation of deprotonated ions, especially in terms of malonyl-glucosyl conjugates, allowing for their proper identification. A typical neutral loss of 248 Da from the precursor ions, corresponding to malonyl-glucosyl moieties, was observed in MS spectra of the unknown isoflavones. Both of the above isoflavone derivatives also gave dominant fragment ions with *m*/*z* 269.0817 and 285.0753 (characteristic of **F** and **B** aglycone structures), which were generated from [M + H]^+^ fragment ions (at *m*/*z* 517.1336 and 533.1309), relating to **F-Gl** and **B-Gl**, respectively. The similarities of fragmentation patterns and characteristic UV spectra to those found for the corresponding glucosides allowed the tentative identification of the unknown isoflavone derivatives as formononetin-7-*O*-glucoside-6″-*O*-malonate (**F-Gl-Mal**) and biochanin A-7-*O*-glucoside-6″-*O*-malonate (**B-Gl-Mal**)—see Figure 2.

#### 2.2.2. Quantitative Profiling of Isoflavones in TML Using the RP-LC/PDA Method

A key step in the phytochemical standardization of TML was the quantitative analysis of isoflavones carried out using the newly developed and validated RP-LC-PDA protocol. The satisfactory resolution of all compounds examined (shown in Figure 1) enabled precise integration of peak areas and subsequent accurate determination of individual isoflavones present in TML. Linear regression curves, obtained from the dependence of peak areas on increasing concentrations (*n* = 6) of prepared methanolic solutions of reference isoflavones, and the resulting equations (y = *a*x + *b*) enabled accurate quantification of the corresponding compounds in TML. High correlation coefficients (*R^2^*) were obtained for all calibration curves, ranging from 0.9994 to 0.9998 (Appendix A). In terms of method sensitivity, satisfactory values for limit of detection (LOD) and quantification (LOQ) were also documented (Appendix A). The results of injection precision determinations for isoflavone standard solutions, expressed as relative standard deviation (RSD, *%*), ranged from 0.3 to 1.8. Regarding the isoflavone content of the TML samples, the evaluation was performed for the combined results of intra- and inter-day determinations made on three consecutive days and repeated in triplicate on each day of the study. The mean RSD values, calculated for the eight individual isoflavones, ranged from 0.2 to 4.8 (*n* = 3) and from 0.5 to 4.1% (*n* = 9) for the intra- and inter-day precision, respectively (Appendix A).

As regards the quantitative results relating to the eight isoflavone components identified in TML, attention was drawn to the predominance of malonyl derivatives of ononin and sissotrin (Figure 3). In this group, a three-fold higher content of **B-Gl-Mal** (20.63 mg/g dry wt) compared to **F-Gl-Mal** (7.31 mg/g dry wt) was also reported. The presence of the aforementioned malonylglucosides as the predominant forms of isoflavones in TML was quite different compared to our previous results obtained for the similarly prepared dry extract of cultivated zigzag clover [2,23]. With respect to the current experiments performed, we hypothesized that the explanation for this phenomenon is the use of plant material derived from wild zigzag clover and harvested at a specific phase of its growth (full flowering). Under these circumstances, various biotic (presence of natural pests) and abiotic (summer exposure to intense UV radiation) factors may have contributed to characteristic changes in polyphenol metabolism in aboveground parts of this taxon, which affected the qualitative profile and content of individual isoflavones. Similar findings for different soybean varieties were published by Akitha Devi et al. [28], who found significant differences in isoflavone content strongly correlated with plant-growth stage. 

In addition to the dominant concentration of malonylglycosyl isoflavone conjugates in TML, the other specific feature of this herbal preparation was the relatively low content of free isoflavone aglycones, namely, biochanin A and formonononetin, as their amounts did not exceed 0.36% and 0.29% (calculated for a dry extract mass), respectively. The total isoflavone content was slightly higher than 5% of the dry extract weight (Figure 3). Considering the percentage of the three main groups of isoflavones, glycosidic and malonylglycosidic derivatives constituted more than 83% of all isoflavone compounds determined in TML. **B-Gl-Mal** and **F-Gl-Mal** were the fractions with the highest concentration (~54.3%), followed by their corresponding glucosides (**B-Gl** and **F-Gl**, respectively) which accounted for about 29.1% of total isoflavones in TML (Figure 4).

The results of the quantitative profiling shown in Figure 3 and Figure 4 are published for the first time for a standardized dried herbal preparation obtained from the aerial parts of wild zigzag clover harvested at full blossoming stage. 

### 2.3. Antioxidant and Antiradical Activity versus Chemopreventive Potential of TML 

Flavonoids and their isoforms (3-phenyl-chromone derivatives), called isoflavonoids, are polyphenolic plant constituents that exhibit antioxidant properties due to the presence of some functional groups and intra-ring molecular structure components (double bonds) capable of interaction and influencing the oxidation state of other compounds. Studies on structure–activity relationships of flavonols showed that their antioxidant and antiradical properties were strongly related to the presence of pyrogallol or catechol moieties in the B-ring, along with the 3-hydroxyl substituent in the unsaturated C ring. Therefore, they were recognized as strong antioxidant and antiradical plant components [29]. Isoflavones (a subclass of isoflavonoids) are predominantly widespread in the family Fabaceae. In this group, formononetin and biochanin A and their glycosyl and malonylglycosyl derivatives were identified in some taxa of the *Trifolium* L. genus [17]. Considering the chemical structure of isoflavones, they showed rather moderate antioxidant activity compared to flavonols [30]. It was documented that, during inactivating reactive oxygen species (ROS), the 2,3-double bond in combination with the 4-oxo group (see Figure 2) was responsible for electron delocalization in the chromone B ring and the formation of another conjugated double bond in the pyran structure [31]. The substitution of hydroxyl groups at positions C-5 and C-7 in the A ring also enhanced antioxidant properties of isoflavones. In addition, the 4-oxo group, together with the 5-hydroxyl moiety, provided an important chelating site for transition metal ions involved in the Fenton reaction. As a general rule, the greater the number of hydroxyl groups in the flavonoid structure, the higher the antioxidant capacity that was documented. Glucosylation or methylation of these groups always led to a significant decrease in the antioxidant activity of flavonoid compounds [32,33].

In our study, a set of four in vitro bioassays was first employed for zigzag clover and the herbal extract prepared thereof to investigate the antioxidant and antiradical potential of TML. For this purpose, experiments were carried out using spectrophotometric (Vis) methods and specific derivatizing reagents. The total phenolic content (TPC) in the zigzag clover lyophilisate was determined using a method based on the reaction with a Folin-Ciocalteu reagent (FCR). The results obtained (~11.0%), calculated as gallic acid equivalents (GAE) per gram of dried herbal preparation and expressed as a percentage, showed that TPC was about twice as high as the total isoflavone content (TIC) in TML documented in phytochemical standardization (Table 1). This would imply that within the group of polyphenols present in TML, isoflavone compounds could play a key role in antioxidant/chemopreventive mechanisms. Our hypothesis was further strengthened by the results of the determination of polyphenolic antioxidants in the CUPRAC method based on the ability to reduce copper (II) ions. The resulting value (shown in Table 1) calculated as a percentage was ~5.0%, giving a nearly 100% correlation with the TIC percentage previously obtained for TML. In addition, the results of the CUPRAC assay may point to phenolic compounds present in TML as potential metal ion chelators that are involved in ROS generation. 

In the second stage of the experiments conducted, the ability of TML to quench artificially generated cation (ABTS^•+^) and anion (DPPH^•^) radicals was evaluated. We documented similar antiradical potency of the TML polyphenolic components in both the DPPH^•^ and ABTS^•+^ assays showing inhibitory concentration (IC_50_) values of 30.18 and 30.30 μg/mL, respectively. Compared to the reference substances (Trolox and gallic acid) TML showed rather moderate antiradical potential (Table 1).

According to the published data, the plant part, extraction method, and solvent used had a significant effect on the content of bioactive phytochemicals and thus on the antioxidant capacity of various clover extracts. Kicel et al. [34] documented that methanolic extract obtained from *T. repens* flowers (IC_50_ 72.3 µg/mL) showed higher antioxidant activity against DPPH^•^ radical than that obtained from the leaves (IC_50_ 179.3 µg/mL) due to the higher TPC ratio. Abiotic and biotic stress factors acting on plants during the growing season can also influence the level of accumulation of phenolic antioxidants [35]. Vlaisavljević et al. [36] studied red clover samples at different stages of the growing season and documented that the plant-growth stage is important for the phytochemical profile of its antioxidants. Extracts obtained from the plant material collected at the earliest growth stage were found to be richest in polyphenols, which contributed to their superior antimicrobial and radical scavenging activity. 

In reference to the high content of malonyl derivatives documented in TML, interesting information was presented in the U.S. patent (by Swiss researchers) that addressed the process of obtaining malonyl derivatives of genistein and daidzein by extracting soybean seeds with methanol/ethanol [37]. The authors concluded that both malonylglucosides exhibited significant antioxidant properties, protecting unsaturated fatty acids and vitamins from oxidative destruction, and therefore these compounds have been targeted for use as additives in the manufacture of cosmetic and food products.

The result of our study and the above reports demonstrate that preliminary procedures, including controlled harvesting of plant material and its extraction and proper phytochemical standardization, can lead to high-quality plant extracts from various species belonging to the Fabaceae family, containing rich complexes of isoflavone compounds with antioxidant and potentially chemopreventive properties. At this point, it is necessary to emphasize once again the important role of environmental and agricultural factors, which can significantly affect the content of specialized plant metabolites, including isoflavones. Numerous investigations (presented below) concerning this matter were carried out for red clover, which is a species closely related to *T. medium*. Booth et al. [20] documented seasonal variability in the concentration of isoflavones in inflorescences and whole green shoots of cultivated red clover during the growing season that significantly influenced estrogenic properties of extracts obtained thereof. Lemežienė et al. [21] and Tsao et al. [22] also confirmed the variable content of isoflavones in this taxon depending on the part of plant and the cultivar type. They also indicated the importance of genetic factors in the biosynthesis of these phytoconstituents.

### 2.4. Cytotoxic Activity of TML and Isoflavone Reference Substances in Breast Cancer Cell Lines

We hypothesized that the cytotoxic effects of TML against breast cancer cells might be strongly correlated with the presence of isoflavone compounds. Therefore, two important representatives of this group, namely, formononetin and its 7-*O*-glucoside (ononin) identified in TML, were selected to be used as positive reference substances for in vitro evaluation of the cytotoxic properties. Considering both the estrogenic (SERM effects) and non-estrogenic mechanisms of the cytotoxic activity of isoflavones, two breast cancer cell lines, MCF-7 (ER-positive) and MDA MB-231 (ER-negative), were used as the research model. Cell viability measured by MTT assay after 24 and 48 h of incubation with TML revealed a markedly increasing cytotoxic effect over time in both tumor cell lines (Figure 5A,D and Figure 6A,D). For higher concentrations (500 to 1000 µg/mL) of the extract used in both cancer cell lines, an approximately two-fold higher inhibition of the cell viability was observed after 48 h compared to the control. As for the ER-dependent MCF-7 cell line, an up to three-fold higher inhibitory effect was even documented after the cells were incubated for 48 h and the extract was used at a concentration of 1000 µg/mL (Figure 6A,D). Detailed analysis of the viability inhibition profiles for both cancer cell lines also indicated that there was a dose–effect relationship for the clover extract examined, i.e. the higher the concentration of TML, the stronger the cytotoxic effect observed in relation to the MDA-MB-231 and MCF-7 cell lines (Figure 5A,D and Figure 6A,D, respectively). At the highest concentration used (1000 µg/mL), TML caused a reduction in the viability of breast cancer cells below 50% for both the ER-positive and triple-negative cell lines (Figure 5A,D and Figure 6A,D, respectively). As regards isoflavone reference substances, the cytotoxic effect of formononetin in the MDA-MB-231 cell line was much higher than that in cells of the MCF-7 line (Figure 5B,E and Figure 6B,E, respectively). Higher concentrations of formononetin (25–100 µg/mL) decreased the survival rate of the MDA-MB-231 cell line in a dose- and time-dependent manner (Figure 5B,E). Isoflavone glycoside–ononin showed no significant effects to either cancer cell line after 24 and 48 h of the experiments performed. There was also no increase in the cytotoxic effect at higher concentrations of the standard (Figure 5C,F and Figure 6C,F). 

Results of our studies showed that TML significantly reduced cell growth rate in a dose- and time-dependent manner in both breast cancer cell lines compared to the control. Thus, one may suspect not only an estrogenic but also a non-estrogenic model of the cytostatic effects of TML, which affects the overall chemopreventive properties of this herbal preparation. Similar biological effects were reported by Gründemann et al. [19] for a medicinal product (Menoflavon^®^ extra) containing the standardized red clover extract (RCE). A high concentration (100 μg/mL) of RCE used was responsible for a cytotoxic effect against MCF-7 and MDA-MB-231 cell lines. Proliferation of RCE-treated MCF-7 cells was observed at very low concentrations (100 pg/mL–100 ng/mL), and pre-incubation with an estrogen antagonist ICI 182,780 did not reverse this effect. Therefore, the ER-independent mechanism of RCE on both estrogen-sensitive and -insensitive cell lines was proposed. Reiter et al. [38] examined in vitro the safety of red clover and soy extracts as well as active isolates using a set of cell lines representing different types of hormone-dependent cancers. The results of this investigation showed that the amount of proliferating cancer cells significantly decreased when both extracts were used, and biochanin A (the predominant isoflavone of red clover) turned out to be the most potent antiproliferative agent. Furthermore, coincubation of ER-sensitive MCF-7 cancer cell lines with isoflavones and steroids to initiate/simulate the pre- and post-menopausal conditions observed in women resulted in decreased rather that increased cell proliferation [38]. Moreover, Dulce-María et al. [26] documented that biochanin A and formononetin obtained from *Cicer arietinum* sprouts not only reduced the number of proliferating MDA-MB-231 cells, but also disrupted cell-membrane integrity. In addition, the aforementioned researchers revealed cell shrinkage, loss of cell asymmetry, and nuclear fragmentation, suggesting cell damage and possible apoptosis. The in vivo studies by Yonemoto-Yano et al. [39] using soybean hypocotyls were relevant to our results regarding the quantitative predominance of malonyl glucosyl derivatives of biochanin A and formononetin in TML. These researchers documented that after oral administration of malonyl daidzin (which is the major isoflavone of soybean hippocotyls) to rats, plasma AUC values for the aglycone daidzein were significantly higher than in animals given the isoflavone glucoside (daidzin). They also proved that the intestinal absorption of malonyl derivatives of isoflavones (as polar hydrophilic compounds that dissolve well in water) is much higher than the corresponding isoflavone glycosides, so they pass more efficiently through the cell membrane of enterocytes. Considering the above findings, we concluded that in our experiments performed both on MCF-7 and MDA-MB-231 cell lines a significant reduction (below 50%) in the viability of breast cancer cell lines (Figure 5A,D and Figure 6A,D) using the highest concentration (1000 µg/mL) of TML could not be explained only by the effect shown by formononetin and biochanin A, as the calculated amounts of these two isoflavone aglycones (per 1000 µg of TML used) were 2.9 and 3.6 µg, respectively. Therefore, we hypothesized that malonyl derivatives of ononin and sissotrin, administered at a total dose of ~27.9 µg (per 1000 µg of TML), could also be involved in the ultimate cytotoxic effects observed in both human breast cancer lines.

### 2.5. Screening for Cytotoxicity of TML in Human Normal Fibroblast WS1 Cell Line

An important part of our study was also to check the potential cytotoxic effects of TML and reference isoflavone substances on the human normal fibroblast WS1 cell line. Satisfactory results were obtained, as the extract showed a lower cytotoxic effect on WS1 cells compared with the cytotoxicity observed against breast cancer cell lines at the same concentrations examined (Figure 7A,D).

## 3. Materials and Methods

### 3.1. Chemicals and Reagents

#### 3.1.1. Reagents and Reference Substances Used in Extraction Procedure and Phytochemical Standardization

Solvents used for extraction and phytochemical standardization were of analytical or chromatographic/LC-MS grade and were provided by Avantor Performance Materials (Gliwice, Poland) and J.T. Baker (Gross-Gerau, Germany), respectively. Ultrapure water (resistance of 18.2 MΩ) was obtained from a Direct-Q system (Millipore, Molsheim, France). Isoflavone reference substances (purity ≥ 95%), namely, genistin, genistein, formononetin, biochanin A, ononin, and sissotrin, were purchased from ChromaDex Inc. (Santa Ana, CA, USA). Stock isoflavone standard solutions, prepared at the concentration range of 0.1–0.2 mg/mL, were properly diluted in methanol (LC-grade) and stored at 4–8 °C.

#### 3.1.2. Chemicals Used in the In Vitro Bioassays

The reagents used for antioxidant assays, namely, FCR (Folin–Ciocalteu reagent), DPPH^•^ (2,2-diphenyl-1-picrylhydrazyl), and neocuproine, ABTS^•+^ (2,2′-azinobis-(3-ethylbenzothiazoline-6-sulfonic acid) were obtained from Sigma Aldrich (Steinheim, Germany).

DMSO (dimethylsulfoxide, Avantor Performance Materials Poland S.A., Gliwice, Poland), MTT (thiazolyl blue tetrazolium bromide), DPBS (Dulbecco’s Phosphate-Buffered Saline, modified, without calcium chloride and magnesium chloride, sterile-filtered), DMEM (Dulbecco’s Modified Eagle Medium—high glucose), FBS (fetal bovine serum), and antibiotics solutions, including penicillin, streptomycin, and trypsin solution (Trypsin 0.25% in HBSS, w/0.1% EDTA-Na_2_, w/o Ca^2+^ and Mg^2+^), were provided by Sigma Aldrich (Steinheim, Germany).

### 3.2. Plant Material and Ultrasound-Assisted Extraction (UAE)

The aboveground parts of *T. medium* were collected at a wild site (about 800 square meters) showing geographic coordinates 51°15′34.74″ N (latitude) and 22°29′21.2064″ E (longitude) and located on the border of two western peripheral districts (Sławin/Szerokie) belonging to the city of Lublin (Poland)—see Appendix A. The area is located on brown-earth soils of loess origin on a slope inclining northward. Harvesting was conducted in late July when zigzag clover was in full flowering. The mean values of temperature and precipitation recorded this month at the collection site were about 21 °C and 95 mm, respectively. After collection, a taxon was authenticated by one of the co-authors (G.Z.). A voucher specimen was deposited in the herbarium of the Department of Pharmacognosy (Medical University of Lublin). Plant material (30 g) was dried at a temperature of 35 °C, powdered in a laboratory mill (sieve 0.75 mm), and then extracted twice with 50% (*v*/*v*) ethanol using ultrasound-assisted extraction (UAE) according to the preparative procedure described in the previously published paper [23]. The UAE procedure was repeated three times. Aqueous–ethanolic extracts were combined and concentrated under vacuum and the residue was dissolved in distilled water and vacuum-dried using an Alpha 2–4 LD plus dryer (Christ, Osterode am Harz, Germany). A highly concentrated lyophilisate (TML) with a drug–extract ratio (DER) of 4:1 (*m*/*m*) was further subjected to phytochemical standardization (Section 3.3 and Section 3.4) and bioassays, described below in Section 3.5. 

### 3.3. Phytochemical Standardization of TML Using RP-LC/PDA Method

The qualitative and quantitative analysis of TML samples was carried out using an Agilent Technologies Model 1100 liquid chromatograph (Waldbronn, Germany) equipped with an autosampler and photodiode array detector (PDA) set at 260 nm. The chromatographic separation of isoflavones was performed on an Aquasil C18 stainless steel column (250 mm × 4.6 mm I.D., dp = 5 μm). To obtain sufficient separation of isoflavone components, a gradient elution program for analytes was developed. A binary solvent system was used that consisted of 1 mM H_3_PO_4_ (A) and acetonitrile (B) at a flow rate of 1 mL/min as follows: 0 min/7, 25 min/20, 60 min/35, 80 min/95, and from 80 to 85 min 7% B in A. The post time was set at 10 min. The injection volume was 10 μL. UV spectra of TML phenolics were recorded within the range of 190–400 nm. The identification of isoflavone compounds was carried out by comparing retention times of the peaks obtained and their UV spectra with those of the reference substances. Spectral data acquisition was performed using Agilent ChemStation Rev. A.10.02 software. In terms of quantitative analysis of isoflavones in TML, an external standard method was used. For this purpose, six-point calibration curves were constructed using methanolic solutions (*C* = 0.01 to 0.10 mg/mL) of reference isoflavones. The linearity of calibration curves referring to individual compounds was assessed using regression coefficients (*R*^2^). Due to the lack of commercially available reference substances, the content of malonyl derivatives of ononin and sissotrin was determined using calibration curves constructed for ononin and sissotrin, respectively. In this case, high matches (exceeding 99.96%) of the UV spectra, experimentally determined for these isoflavones and their malonyl conjugates, were also taken into account. Regarding the method precision, both the reference isoflavone substances and the TML methanolic solution (*C* = 5 mg/mL) were prepared in triplicate (*n* = 3), and analyzed within three consecutive days for intra- and inter-day precision, respectively. The data were expressed as relative standard deviation (RSD, %). Detailed information on the validation parameters used in the RP-HPLC method is provided in the Appendix A.

### 3.4. Qualitative Profiling of TML Using RP-LC/PDA/ESI-QTOF/MS-MS Method

Comparative qualitative analysis of TML isoflavones was carried out using an Agilent Technologies system (Santa Clara, CA, USA) consisting of an LC 1290 Infinity chromatograph coupled to a PDA detector and a 6530B QTOF-MS/MS mass spectrometer equipped with an electrospray soft ionization (ESI) source. Chromatographic separation of isoflavone analytes was performed on a Zorbax Stable Bond-C18 narrow-bore column (2.1 mm × 150 mm, dp = 3.5 μm). Volumes of the injected sample aliquots were 10 μL. A mobile-phase gradient (at a flow rate of 0.2 mL/min) composed of acetonitrile (B) and water (A) with 0.1% (*v*/*v*) formic acid and 10 mM ammonium formate (pH~3.5) was employed as follows: 0 min/5, 5 min/15, 25 min/20, 55 min/45, 65 min/95, and 70 min/5% B in A during the remaining 5 min. The column re-conditioning time was 12 min. The mass spectra of compounds examined were recorded in positive ionization mode in the range of 100–1000 *m*/*z* using Agilent MassHunter Workstation software. The collision-induced dissociation (CID) energies were set to 20 and 40 eV to obtain MS/MS spectra for the two precursor ions (in each analysis) possessing the highest intensities. Other optimized ESI source operating parameters were set as described in our previous work [2]. Mass spectrometric confirmation of the molecular structure of the analyzed compounds was conducted on the basis of their fragmentation patterns compared with data recorded for authentic isoflavones.

### 3.5. Antioxidant and Radical Scavenging Assays 

All antioxidant and radical scavenging assays presented below were performed using a Thermo Electron 10S Series UV-Vis Spectrophotometer (Thermo Electron Scientific Instruments, Madison, WI, USA), which provides absorption measurements in the wavelength range of 325–1000 nm using a stable bandwidth of 5 nm (wavelength accuracy of 1 nm).

#### 3.5.1. Determination of Total Polyphenolic Content (TPC)

The total polyphenol content of TML was evaluated according to the method elaborated by Singleton and Rossi [40] using the Folin–Ciocalteu method. One mL of the stock methanolic solution of TML (*C* = 0.5 mg/mL), 5 mL of distilled water, 0.5 mL of the Folin–Ciocalteu reagent (FCR), and 1.5 mL of 20% (*m*/*v*) aqueous solution of Na_2_CO_3_ were added to a volumetric flask (10 mL) and made up to nominal volume with distilled water. The prepared sample was vortexed and afterwards allowed to stand in darkness for 60 min at room temperature. Then, the absorbance (A_sample_) was measured at 765 nm against a blank sample (A_blank_) containing 1 mL of water instead of the lyophilisate. As the TPC was determined as gallic acid equivalent (GAE), methanolic solutions of gallic acid (*C* = 0.05–0.25 mg/mL) were prepared and the six-point calibration curve was constructed following the same aforementioned procedure. The spectrophotometric protocol was performed in triplicate for both the sample examined and the reference substance and repeated over the subsequent three days. TPC was calculated as mg GAE per 1 g lyophilisate (dry weight) using the following equation:GAE = (A_sample_ × 0.1)/(A_blank_) × 0.0005,(1)

#### 3.5.2. Copper Reduction Assay 

The evaluation of both the antioxidant and metal ion-chelating capacities of TML was performed with an copper reduction assay (CUPRAC) according to the method described by Apak et al. [41]. One mL of the stock methanolic solution of TML (*C* = 0.2 mg/mL), 1 mL of ethanolic solution of neocuproine (Nc), 1 mL of ammonium acetate (pH 7), and 1 mL of CuCl_2_ were pipetted into the calibrated flask, vortexed, and filled with 10% (*v*/*v*) ethanol to the final volume (5 mL). After 30 min, the absorbance (A_sample_) of a colored solution was measured at 450 nm against a blank sample (A_blank_). The quantitative results were calculated as gallic acid equivalent (GAE); therefore, stock methanolic solution of gallic acid (*C* = 0.002 mg/mL) was prepared and the six-point calibration curve was constructed following the same aforementioned procedure. The spectrophotometric protocol was performed in triplicate for both the sample examined and the reference substance, and repeated over the subsequent three days. The Cu (II)-to-Cu (I) ion reduction capacity of TML was calculated using the following equation:GAE = (A_sample_ × C_GAE_ × 5)/(A_blank_ × 0.0002),(2)

#### 3.5.3. DPPH^•^ Anion Radical Scavenging Assay 

The DPPH^•^ (2,2-diphenyl-1-picrylhydrazyl) radical scavenging abilities of TML was determined spectrophotometrically as described before by Brand-Williams et al. [42] with slight modifications. An aliquot (10–100 µL) of a stock methanolic TML solution (*C* = 0.5 mg/mL) and corresponding volumes of 10% (*v*/*v*) ethanol to obtain 100 µL were added to a test tube using a microsyringe, followed by 1 mL of a DPPH^•^ solution (*C* = 100 μM) in 99.8% (*v*/*v*) ethanol and 1 mL of 96% (*v*/*v*) ethanol. The content of the test tube was vortexed and incubated (30 min) for color change. The absorbance (A_sample_) was measured at 517 nm against the corresponding blank (A_0_). Additionally, the antioxidant activity (AA) of gallic acid (*C* = 0.05 mg/mL) and Trolox (*C* = 1 mg/mL) were examined using the changing volumes of the reference substance solutions. The percentage of inhibition AA (%) of DPPH^•^ radical species was calculated using the following formula:AA (%) = [(A_0_ − A_sample_)/A_0_] × 100,(3)

The final results are expressed as IC_50_ value (corresponding to the antioxidant/antiradical agent concentration that decreases the initial radical amount by 50%) using linear regression analysis obtained for the curves showing the dependence on concentration versus antioxidant activity. The spectrophotometric protocol was performed in triplicate for both the sample examined and the reference substance and repeated over the subsequent three days.

#### 3.5.4. ABTS^•+^ Cation Radical Scavenging Assay 

The ABTS^•+^ radical cation scavenging assay for TML samples was carried out according to the method described by Gu et al. [43]. In brief, free radical cations were generated through the oxidation of ABTS (2,2-azinobis-(3-ethylbenzothiazoline-6-sulfonic acid)) water solution (7 mM) by potassium persulfate solution (2.45 mM). The reaction was performed for 12 h in the darkness. Directly before use, the absorbance of the dark purple-colored ABTS^•+^ solution (measured at 734 nm) was adjusted to the desired value of about 0.700 (±0.020) by adding 96% (*v*/*v*) ethanol, thus obtaining the working solution. Then, 10–100 µL of TML samples (**C* =* 2.0 mg/mL) were added with a microsyringe to a calibrated flask and filled up to a volume of 5 mL using the working cation radical solution. The blue–green color disappearance in the sample examined (A_sample_) was measured at 734 nm after vortexing (1 min) against 50% ethanol. A similar procedure was performed for the blank sample (A_0_). Additionally, the antioxidant activity (AA) of gallic acid (*C* = 0.1 mg/mL) and Trolox (*C* = 0.2 mg/mL) was examined using the changing volumes of the reference substance solutions. The results, showing the percentage of inhibition AA (%) of ABTS^•+^ cation radical, were calculated as follows:AA (%) = [(A_0_ − A_sample_)/A_0_] × 100,(4)

The final results are expressed as IC_50_ ratio (corresponding to the antioxidant/antiradical agent concentration that decreases the initial radical amount by 50%) using linear regression analysis obtained for the curves showing the dependence on concentration versus antioxidant activity. The spectrophotometric protocol was performed in triplicate for both the sample examined and the reference substance and repeated over the subsequent three days.

### 3.6. Cytotoxicity Screening Assay 

#### 3.6.1. Cell Culture

MCF-7 (progesterone and estrogen receptor-positive human breast epithelial adenocarcinoma) and MDA-MB-231 (a triple-negative human breast adenocarcinoma) cell lines were obtained from American Type Culture Collection (Manassas, VA, USA) and cultured using DMEM—high glucose (Sigma Aldrich, Steinheim, Germany) supplemented with 10% fetal bovine serum (FBS, Sigma Aldrich, Steinheim, Germany), 100 U/mL of penicillin, and 100 mg/mL of streptomycin (PenStrep, Sigma Aldrich, Steinheim, Germany). Cells were incubated at 37 °C in a humidified atmosphere of 5% CO_2_. The WS1 cell line was maintained in Eagle’s Minimum Essential Medium (EMEM) supplemented with 10% fetal bovine serum, 100 U/mL of penicillin, and 100 mg/mL of streptomycin. Cell incubation was performed at 37 °C in a humidified atmosphere of 5% CO_2_.

#### 3.6.2. MTT Assay Protocol

Cytotoxic effects of TML and isoflavone reference substances (i.e., formononetin and ononin) were examined on MCF-7 and MDA-MB-231 human breast cancer cells and WS1 cells using an MTT assay. Stock solutions of the tested compounds were prepared via dissolving in a sterile DMSO. The suspension of cells was prepared at a density of 1 × 10^5^ cells/mL and then transferred to 96-well cell culture plates (NUNC, Roskilde, Denmark). After 24 h of incubation, the medium was removed from each well and then cells were incubated for the next 24 h/48 h with different concentrations of the compounds examined. The cytotoxic effect of TML and isoflavones was assessed using the MTT assay. After the respective incubation period (i.e., 24 h or 48 h), MTT solution (5 mg/mL) was added to all wells, and the plates were incubated for 3 h. In the next step, 100 µL of 10% SDS buffer solution per well was added and after an overnight incubation the absorbance was measured at 570 nm using a microplate reader (Epoch, BioTek Instruments Inc., Winooski, VT, USA). 

### 3.7. Statistical Analysis 

Statistical evaluation of the quantitative results obtained for individual isoflavones during phytochemical standardization and antioxidant/antiradical assays involved verifying the null hypothesis (H_0_), which assumed equality of variance across statistical series and no differences between group means, analyzed for three series (*n* = 3, in each) in the intra- and inter-day precision tests. The statistics were calculated in the GraphPad Prism 5 program (GraphPad Software, San Diego, CA, USA) using the *F*-Snedecor test in the one-way analysis of variance (ANOVA) test. Statistical significance was set at *p* < 0.05.

For the MTT assay, the viability of the treated MCF-7, MDA-MB 231, and WS1 cells was expressed as % of the viability of control (untreated) cells. Each experiment was repeated thrice. The results are expressed as means ± SD. Data were analyzed using GraphPad Prism software (version 5.01). The statistical significance among the groups was determined using ANOVA analysis with Tukey’s *post-hoc* test. 

## 4. Conclusions

This study provides the results of new and original research on the biological (antioxidant, chemopreventive, and cytostatic) effects of isoflavone components identified and quantified in a vacuum-dried extract (TML) obtained from the flowering aboveground parts of wild zigzag clover. In the introduction, we highlighted the dual nature of isoflavones (including biochanin A and formonononetin conjugates found in red and zigzag clover) being both polyphenolic antioxidants and selective modulators of estrogen receptors, which determines their potential for complex and multidirectional interactions at the cellular and organ level. We documented, for the first time, the presence of malonylglycoside derivatives of biochanin A and formononetin in TML as dominant compounds that contributed more than 54% to the total content of isoflavones determined in this herbal product. This is also the first report on the noticeable cytotoxic effects of TML, performed on the MCF-7 (estrogen-dependent) and MDA-MB-231 (estrogen-independent) human breast cancer lines. Based on the literature data and the antioxidant/antiradical studies performed, we hypothesized that, in addition to the isoflavone aglycones (formononetin and biochanin A), their malonylglycosyl derivatives could have a significant impact on the chemopreventive and cytotoxic effects of TML against both human breast cancer cells. The results of this study suggest that zigzag clover may be regarded as a rich botanical reservoir of isoflavone antioxidants and selective ER modulators that exhibit chemopreventive effects, and as an alternative to red clover, currently widely used for the prevention and treatment of various ailments in menopausal women.

## Figures and Tables

**Figure 1 pharmaceuticals-15-00699-f001:**
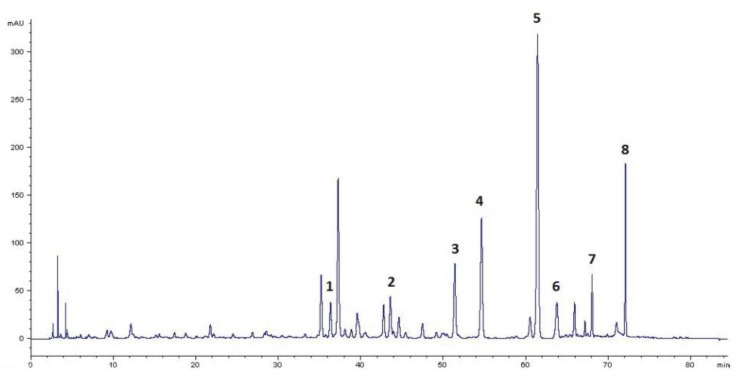
RP-LC/PDA chromatogram (recorded at 260 nm) of major isoflavone constituents found in TML. Peak numbers: **1**—genistin; **2**—ononin; **3**—ononin malonate; **4**—sissotrin; **5**—sissotrin malonate; **6**—genistein; **7**—formononetin; **8**—biochanin A.

**Figure 2 pharmaceuticals-15-00699-f002:**
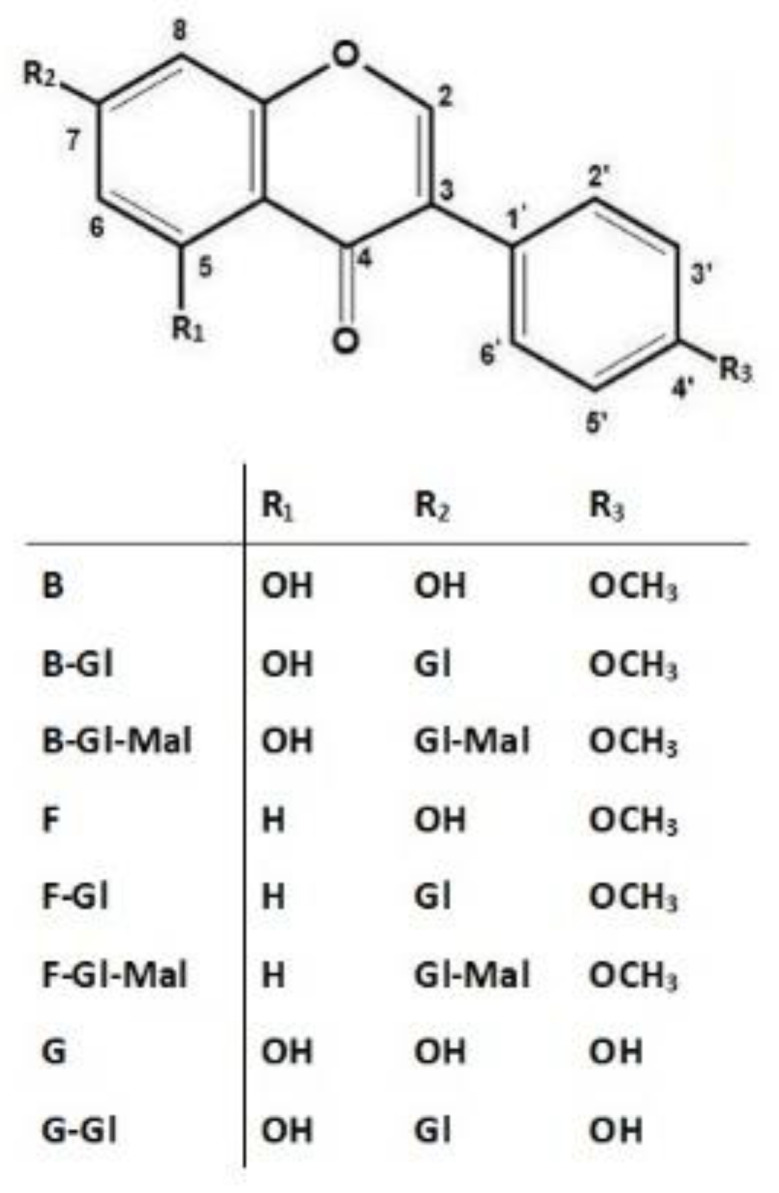
Chemical structure of isoflavones identified in TML. Explanation of compound name abbreviations: G—genistein; G-Gl—genistein-7-*O*-glucoside (genistin); F—formononetin; F-Gl—formononetin-7-*O*-glucoside (ononin); F-Gl-Mal—formononetin-7-*O*-glucoside-6″-*O*-malonate; B—biochanin A; B-Gl—biochanin A-7-*O*-glucoside (sissotrin); B-Gl-Mal—biochanin A-7-*O*-glucoside-6″-*O*-malonate.

**Figure 3 pharmaceuticals-15-00699-f003:**
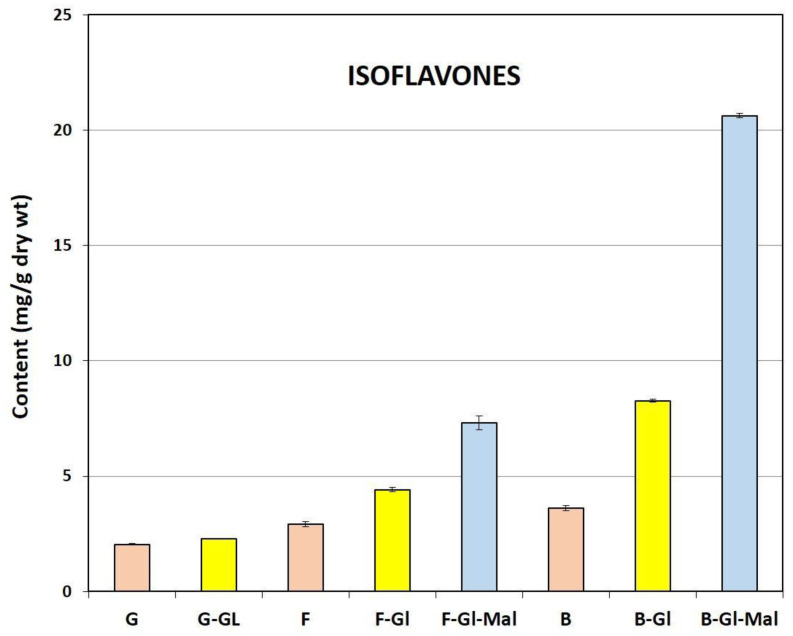
The mean (*n* = 9) content (mg/g dry wt) of isoflavones determined in TML. Compound name abbreviations as in Figure 2. The colors of each graph refer to the three isoflavone groups shown in Figure 4.

**Figure 4 pharmaceuticals-15-00699-f004:**
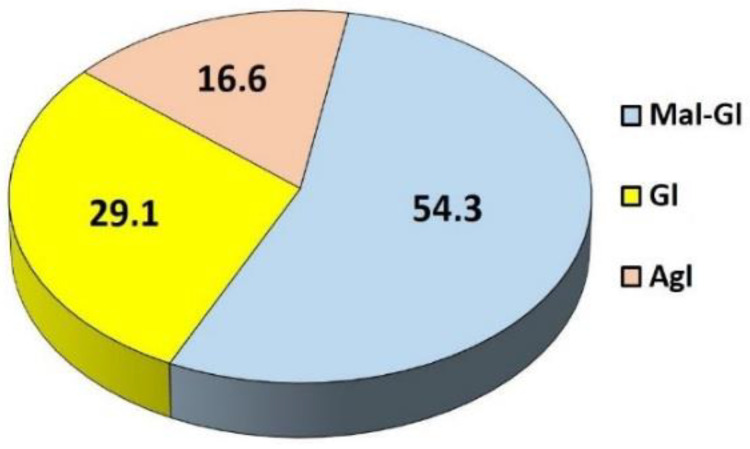
The average (*n* = 9) percentage of isoflavone aglycones (**Agl**) and their derivatives, including glycosides (**Gl**) and malonylglycosides (**Mal-Gl**), per total isoflavone content determined in TML.

**Figure 5 pharmaceuticals-15-00699-f005:**
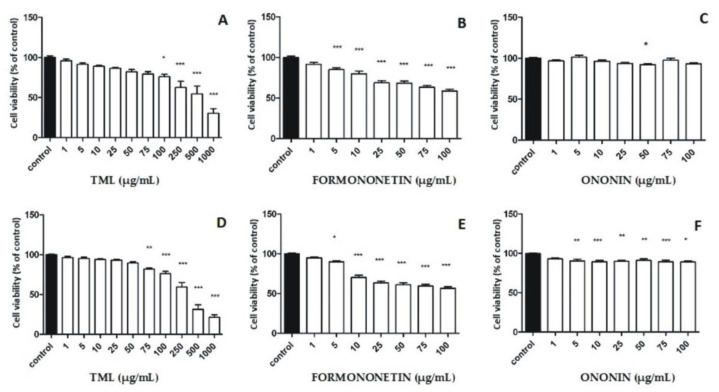
Effect of isoflavone reference substances and TML on the viability of MDA-MB-231 human breast cancer cells after 24 h (**A**–**C**) and 48 h (**D**–**F**) incubation; results expressed as percentage of relative viability of treated cells compared to control. Values in graphs shown as mean (*n* = 3) ± SD. Statistical differences * *p* < 0.05, ** *p* < 0.01, *** *p* < 0.001 for control versus cell lines incubated with TML or a reference substance were determined using one-way ANOVA analysis with Tukey’s post-hoc test.

**Figure 6 pharmaceuticals-15-00699-f006:**
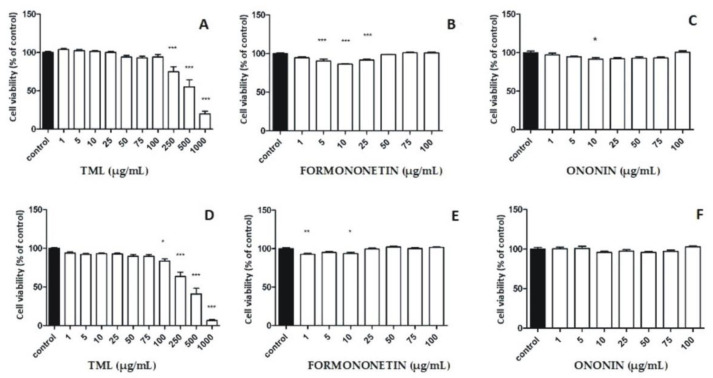
Effect of isoflavone reference substances and TML on the viability of MCF-7 human breast cancer cells after 24 h (**A**–**C**) and 48 h (**D**–**F**) incubation; results expressed as percentage of relative viability of treated cells compared to control. Values in graphs shown as mean (*n* = 3) ± SD. Statistical differences * *p* < 0.05, ** *p* < 0.01, *** *p* < 0.001 for control versus cell lines incubated with TML or a reference substance were determined using one-way ANOVA analysis with Tukey’s post-hoc test.

**Figure 7 pharmaceuticals-15-00699-f007:**
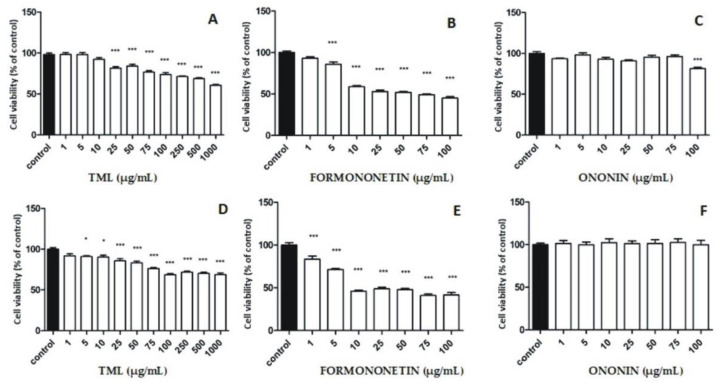
Effect of isoflavone reference substances and TML on the viability of normal skin fibroblast cells (WS1) after 24 h (**A**–**C**) and 48 h (**D**–**F**) incubation; results expressed as percentage of relative viability of treated cells compared to control. Values in graphs shown as mean (*n* = 3) ± SD. Statistical differences * *p* < 0.05, *** *p* < 0.001 for control versus cell lines incubated with TML or a reference substance were determined using one-way ANOVA analysis with Tukey’s post-hoc test.

**Table 1 pharmaceuticals-15-00699-t001:** The content of polyphenolic antioxidants in TML determined in the Folin–Ciocalteu (FC) and CUPRAC methods, and their antiradical properties evaluated in ABTS^•+^ and DPPH^•^ scavenging assays.

FC	CUPRAC	ABTS^•+^	DPPH^•^
(mg GAE/g dry wt)	(mg GAE/g dry wt)	IC_50_ (μg/mL)	IC_50_ (μg/mL)
107.50 ± 0.26	48.00 ± 1.87	30.30 ± 0.18 2.75 ^a^ ± 0.07 0.69 ^b^ ± 0.004	30.18 ± 0.37 3.00 ^a^ ± 0.90 0.89 ^b^ ± 1.80

All results shown in Table 1 are expressed as mean values ± SD of nine independent measurements. Abbreviations: ^a^—Trolox (reference substance), ^b^—gallic acid (reference substance); GAE—gallic acid equivalent.

## Data Availability

Data is contained within article.

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
