# Peer review of "In Vitro Evaluation of the Antioxidant Activity and Chemopreventive Potential in Human Breast Cancer Cell Lines of the Standardized Extract Obtained from the Aerial Parts of Zigzag Clover (Trifolium medium L.)"

_pharmaceuticals, 2022, doi:10.3390/ph15060699_

Round 1

Reviewer 1 Report

In the abstract authors should define the area considered for the study (the name of the sampling area should be added) and they should add little information (2-3 lines) on the properties of Trifolium medium L. or at least the reason for which this plant was analyzed. Please reduce some keywords.

Too long introduction about the state of the art of the plant biological properties (reduce the number of references); authors should focus on the main research topics and relevant questions should be addressed. Information related to phytochemicals in Trifolium medium L. and relative biological properties should be slightly reduced, while information on the influence of agro-environmental conditions on the concentration of molecules should be added. Literature reported that quali-quantitative amounts of bioactive compounds often depend on genotype, pedoclimatic conditions, phenology of plants, etc.

In Material&Methods more information on raw materials should be added (information on sampling site and time, pedoclimatic conditions, genotype, etc). Did the authors use validated HPLC methods? Please add a Table (also in Supplementary Materials) with validation data or at least calibration parameters.

In the result/discussion section, authors should compare their results with other species used for similar purposes (same phytochemicals?).

The conclusion is clear in relation to the study, but it should be linked in a better way to the other parts of the paper. Delete redundant information, please.

Reviewer 2 Report

Comments

The manuscript can be of interest to wide readers of Pharmaceuticals Journals and contributes to existing knowledge on the subject matter. However, I have pointed out a few pertinent points for improving the clarity of the content and boosting the scientific soundness of the manuscript.

Introduction

Write the distribution, medicinal values, different biological activities, and important phytochemicals of the experimental plant used in the experiment, citing important previous work and references.

Give citations to lines 32 – 37.

Give citations to lines 61 – 65

Give citations to lines 70 – 74

The research hypothesis is missing

Results and Discussion

This is well written, however Authors need to strengthen the discussion section by adding more interpretations of recorded findings supported by peer-findings

Line 1224-133: Adjust these lines in the material and methods section

Change the caption of Table 1

“Table 1. Antioxidant and antiradical activity of TML and reference substances evaluated using 302 the Folin-Ciocalteu (FC), CUPRAC, ABTS•+ and DPPH• methods.”

Folin-Ciocalteu (FC), is not for measuring Antioxidant and antiradical activity.

TML and reference substances.

“Table 1. Name the references substances

“Table 1. Why FC and CUPRAC values are incomplete. ABTS and DPPH were tested for three samples.

Give references for the following

Line 320: “Similarly, plant growth stage was reported to be important for the phytochemical profile of clover antioxidants”.

Line 396: Biochanin A, the predominant isoflavone of red clover, was  reported to be the most potent antiproliferative agent.

.

Line 400: Moreover, in another study biochanin A and formononetin  obtained from Cicer arietinum sprouts, not only reduced the number of proliferating MDA-MB-231 cells, but also disrupted cell membrane integrity.

Line 406-412: Give references

Figure 2, 5,6,7 are not clear. Provide clear pictures

Material and methods

Information on study coordinates must be added, along with geographical map drawing

Round 2

Reviewer 2 Report

The author of the manuscript has revised and corrected the article carefully and responded to all the queries. Now the paper can be accepted in its present form.